# Theoretical Basis and Technical Method of Permeability Enhancement of Tectonic Coal Seam by High Intensity Acoustic Wave In Situ

Weidong Li [1], Yongmin Zhang [2,*], Dalong Wang [3], Cunqiang Chen [3], Yongyuan Li [1], Youzhi Zhao [2], Shuo Zhang [2], Jing Ren [2] and Yong Qin [4]

1   Huaneng Coal Technology Research Co., Ltd., Beijing 100071, China
2   State Key Laboratory of Electrical Insulation for Power Equipment, Xi'an Jiaotong University, Xi'an 710049, China
3   Mining Branch of Huaneng Yunnan Diandong Energy Co., Ltd., Qujing 655500, China
4   Key Laboratory of Coalbed Methane Resources and Reservoir-Forming Process, China University of Mining and Technology, Ministry of Education, Xuzhou 221008, China
*   Correspondence: hpeb2006@126.com

**Abstract:** Tectonic coal seams are characterized by soft, low permeability and high gas outburst. The traditional gas control method is the intensive drilling and extraction in this seam, which is not only large in engineering quantity, high in cost, difficult to form holes and low in extraction efficiency, but also easy to induce coal and gas outburst, which is a difficult problem for global coal mine gas control. To solve this difficult problem, the controllable shockwave equipment developed by the author's team and successfully applied in the practice of permeability enhancement of coal seam, combined with the principles of shock vibration sound wave generation and shock wave attenuation and evolution in the rock stratum, a new idea of loading a controllable shock wave in the roof and floor of coal seam is proposed. The shock wave first attenuates and evolves into a high-strength sound wave in the roof and floor rock stratum, and then enters and loads into the coal seam to achieve the purpose of increasing permeability without damaging the physical properties of the tectonic coal seam and facilitating the opening of the original fractures. According to the new technical ideas, the implementation scheme and key parameters of the gas pre-extraction models in tectonic coal seam are designed, including the penetration drilling, roof and floor horizontal holes, shield tunneling and the high-strength acoustic wave of the working face, which provides a new technical approach to solve the problem of high efficiency and low cost gas extraction in the tectonic coal seam.

**Keywords:** high-strength acoustic wave; controllable shockwave; structurally controlled coal bed; coal seam permeability enhancement; coal seam fissure; interlayer drilling

## 1. Introduction

Tectonic coal seams are the result of strong geological compression and shearing deformation during geological history, and are characterized by low cohesion, low strength, low elastic modulus and low permeability [1]. Most coal basins in China have experienced multiple periods of structural movement, and tectonic coal seams are widely developed [2,3]. Most coal and gas outburst accidents worldwide are related to tectonic coal seams [4].

Compared with primary structure coal seam, constructing extraction boreholes in a tectonic coal seam is more difficult [5]. Meanwhile, tectonic coal seams generally have a higher adsorption and desorption capacity and diffusion coefficients [2,6,7], coupled with extremely low mechanical strength, which leads to a great tendency of coal and gas outburst. Coal and gas outburst often cause problems, such as water blockage, gas lock and coal powder blockage during gas extraction, which is difficult to recover once damaged, especially for tectonic coal seams.

Previous researchers have conducted many explorations on the construction of coal seam gas drilling and extraction technology, but drilling construction and permeability enhancement operations are either limited to the coal seam itself [8–12], or require the construction of horizontal wells on the roof from the ground [13]. China's coal industry has tried a variety of coal penetration methods and carried out field tests, such as the impact method, vibration method and acoustic method, etc. [14,15], to improve the permeability of structural coal seam and improve the extraction effect. Among them, the high-frequency acoustic wave experiment explains the mechanism of the permeability enhancement of the coal seam, gives the basic parameters and verifies the feasibility from the simulation experiment level.

As for high-strength acoustic wave, as early as the 1950s, the former Soviet Union and the United States have successively carried out relatively mature research on acoustic oil recovery technology. Russian scientists pointed out that when the sound field intensity is greater than 1 kw/m$^2$, it can make the thermal and qualitative changes of the oil layer, and then peel off the oil layer adsorbed in the rock. According to the measured acoustic parameters in the oil layer of the Henan Oilfield in China, the shock wave generated by the hydroelectric effect with 5 kJ energy in the oil layer is 260 m, and the acoustic intensity below 200 Hz is still 150 db. The Xian Xuefu academician team of Chongqing University, the China University of Mining and Technology and Xi'an Jiaotong University have successively carried out tests and practices on the acoustic field in the aspects of reservoir plugging removal and coal seam permeability enhancement. The results show that when the acoustic pressure is greater than the anti-swelling and shear strength of the coal seam, the existing pores and fractures in the coal seam will be torn in a shear tensile mode.

However, due to the restriction of the principle of vibration acoustic wave, the sound intensity and the sound frequency are related, and improving the output sound intensity will inevitably increase the sound frequency, while the high frequency acoustic wave will rapidly attenuate in the coal seam, resulting in the limited range of permeability enhancement. Therefore, there was no research and development of the equipment for enhancing the coal seam permeability with high intensity acoustic wave in the early stage. Except for the results of the author's team, there are no reports on the high-strength sound waves with controllable shock waves as the source.

In this paper, the author's team proposed controllable shock wave technology that implemented shock in the coal seam and its surrounding roof and floor, using the stratum to convert the shock into high-strength sound wave acting on the coal seam, and repeatedly exciting the coal seam with multiple shocks. The controllable shock wave technology has the advantage of causing "no damage to the coal seam". According to the physical characteristics of the target coal seam, the shock wave parameters, operation times, operation points and other parameters are adjusted to achieve the effect of expanding and communicating the coal seam fractures, without causing macro cracks, and to protect the coal seam structure from damage to the greatest extent. In addition, the controllable shock wave does not need to inject any other material into the coal seam, which is different from coal seam reconstruction technology, such as hydraulic fracturing [16].

The author's team has conducted extensive research and trials, based on the Griffith crack theory of brittle medium and the mechanical properties of coal seams. The stress of the original crack and the corresponding sound pressure and sound intensity of the high-strength acoustic wave propagating coal seams were analyzed and calculated and compared with the existing experimental results, the strength window of the high-strength acoustic wave propagating coal seams was determined, i.e., the difference between the tensile and shear strength of coal seams and the expansion stress strength of the coal seams. This paper expounds the theoretical basis of high-strength acoustic wave antireflection in structural coal seams, completes the design of parameters, layout of through hole operation, layout of horizontal hole operation in the heading face, layout of the drilling

operation in the roof of the trench and predicts the effective range of high-strength acoustic wave antireflection.

On the basis of the successful application of shockwaves to enhance the permeability of harder coal seams, our team has been devoted to the theoretical exploration and field experiments of controllable shockwaves to enhance the permeability of tectonic coal seams. We aim to explore new technical approaches to solve the problem of efficient, low-cost and environmentally friendly gas extraction from tectonic coal seams.

## 2. Theoretical Basis of High-Strength Acoustic Penetration of Tectonic Coal Seam

Coal seam is both the object of acoustic wave action and the medium for propagating acoustic wave, and acoustic permeability enhancement of coal seams is the cleanest physical method of permeability enhancement. The fundamental concept of using sound waves to enhance permeability in structural coal seams is to expand the pre-existing fractures and create a network of fractures, which increases coal seam porosity. At the same time, the high-frequency vibrational shear force generated at the adsorption interface is utilized to promote gas desorption.

### 2.1. The Attenuation Law of Shock Wave to High-Strength Sound Wave

At the near-field of a shock wave, it manifests as a shock tension causing material to fracture; then, the shock wave attenuates into a compression wave, utilizing compression, tension and shear forces to tear apart the material; finally, the shock wave attenuates into an elastic sound wave, where, if the material's rupture strength is lower than the intensity of the elastic wave, no new cracks are formed under the effect of elastic waves, but the pre-existing cracks are extended using the stress concentration characteristics at the crack front. According to this principle and according to the characteristics of structural coal and increasing demand, the shock wave operation from the coal seam to coal seam roof or floor, after the rock attenuation evolution of high-strength acoustic input coal seam, achieve the purpose of porous coal seam, elastic wave high frequency vibration decoupling adsorption at the same time to promote the role of gas desorption (Figure 1).

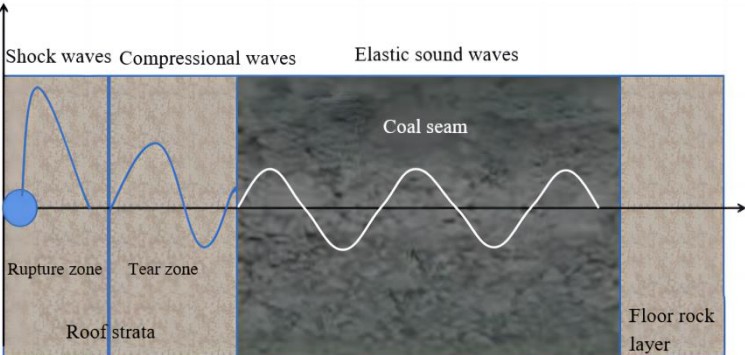

**Figure 1.** Schematic diagram of shock wave attenuation and tectonic coal seam permeability enhancement with high intensity acoustic wave in situ.

### 2.2. Principle of High-Strength Acoustic Wave Coal Seam Penetration

(1)    Technical conditions of high-strength acoustic wave penetration of coal seams

The premise of using high-intensity acoustic waves to increase the penetration of a tectonic coal seam is to produce an extension of the original fractures in the seam without creating new cracks and thus without damaging the macrostructure of the seam. To achieve this goal, three requirements need to be met. Firstly, the sound pressure should be less than or equal to the coal seam's tensile and shear strength. Secondly, the sound pressure should exceed the stress threshold of the pre-existing cracks in the torn coal seam, thereby expanding the cracks in the coal seam. Thirdly, the effective permeable area should be maximized, and the effective permeable radius should be expanded as much as possible.

This approach can increase the gas permeability of the coal seam without causing a coal and gas outburst. The effective area of the high-intensity acoustic wave enhancement of coal seam permeability is defined as a spherical area centered on the sound source. The radius of the effective area is equal to the distance until the sound pressure attenuates to the stress threshold required to tear the coal seam.

(2)  Mechanical basis of the permeable structural coal seam

The compressive strength and elastic modulus of different structural coal under different surrounding pressures were analyzed [17–23]. The uniaxial compressive strength of structural coal is generally less than 3 MPa, and the elastic modulus of most structural coal is less than 1 GPa, which is far lower than that of the native structural coal (Figure 2). The structural coal seam with such low mechanical parameters brings great difficulties to the technical selection of the coal seam by mechanical measures. When the loading is strong enough, it will damage the coal seam structure and cause the opposite effect; when the loading strength is not enough, the penetration effect is not achieved. The test results of coal samples from Qinan Mine in Huaibei Coal field show that the effective elastic modulus and tensile strength of both structural coal and primary structural coal decreased exponentially with an increasing particle size, but the values of both parameters of primary structural coal were significantly larger than those of the same particle size conditions. (Figure 3) [24].

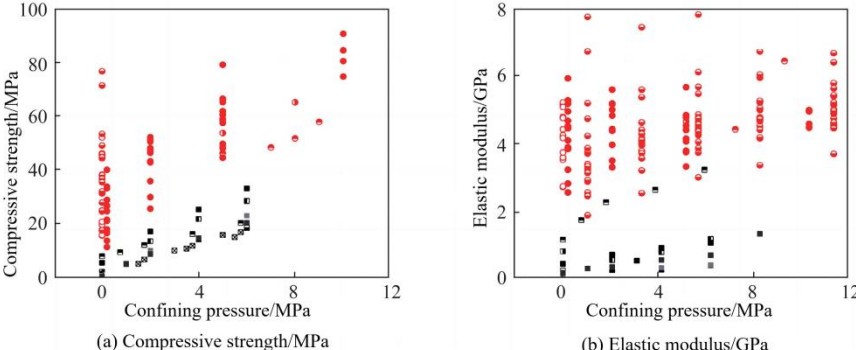

(a) Compressive strength/MPa   (b) Elastic modulus/GPa

**Figure 2.** Difference of mechanical properties between primary structure and tectonic coals; □~■ Structural coal, data mainly from Huainan, Guizhou and Songzao mining areas in China (represented by different patterns); ○~● Primary coal, mainly from China's Qianqiu, Tangshan, Yunnan, Pingdingshan, Ordos and 11 South America (in different patterns).

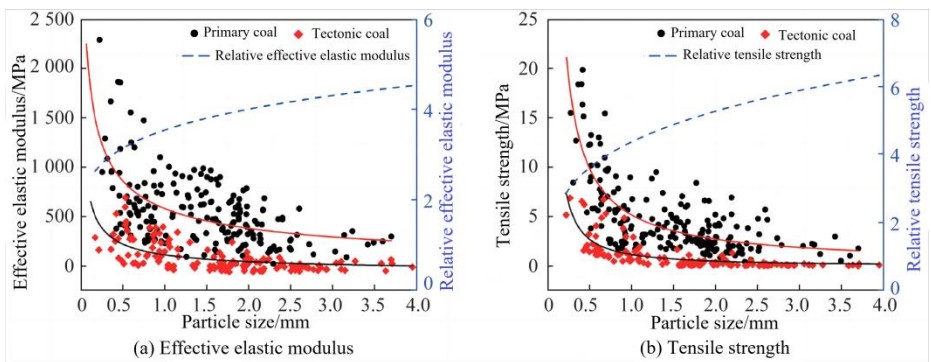

(a) Effective elastic modulus   (b) Tensile strength

**Figure 3.** Relationship between mechanical properties of primary structure and tectonic coals with particle size.

The acoustic strength of the coal seam should be lower than the tensile and shear strength of coal seam. Therefore, we believe that the sound pressure value of the high-strength sound wave loaded into the structural coal should be lower than 1 MPa.

(3)  Effective penetration increase range

One of the challenges in enhancing the permeability of coal seams in structural coal seams is that various drilling and excavation measures have a limited effect on the permeability zone of the coal seam, with an effective radius of approximately 2 m.

The method of protective layer unloading pressure and increasing coal seam is essentially large area balanced unloading, so it is successful. The gas extraction effect of a coal mine in northwest China is shown in Figure 4. According to the calculation of the distance between the working face and the borehole during the active gas period, the effective penetration area formed by the pressure discharge of the protective layer reaches 100 m. Expanding the limited penetration area can double the penetration enhancement effect. When the shock wave operation point is set in the roof and bottom slab layer of the coal seam, the shock wave propagation distance is far greater than the propagation distance set at the coal seam, which can expand the area of effective sound wave entering the coal seam and increase the effective penetration range.

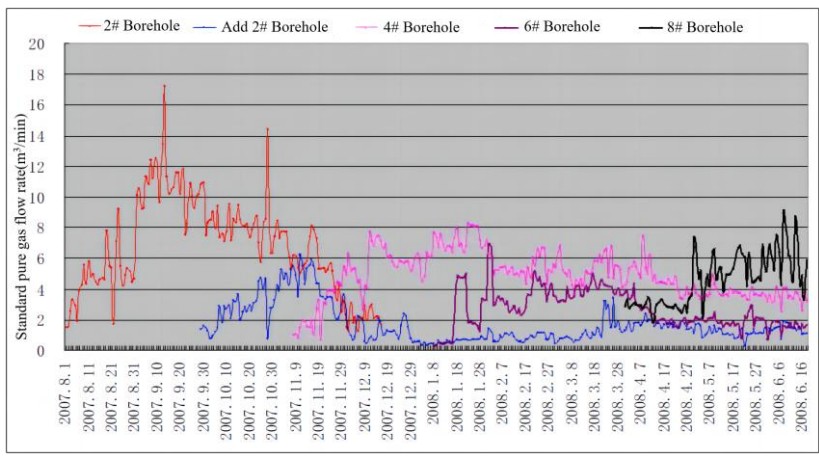

**Figure 4.** Comparison of standard net flow rate of gas extraction from boreholes on wind lane side of a coal mine.

(4)  The relationship between acoustic wave intensity and coal seam crack extended stress

There are cracks in the actual material. When the applied stress is very low, the local stress at the crack tip increases due to stress concentration. When the stress reaches the theoretical breaking strength, $\sigma_c$, the crack propagates and a brittle fracture occurs. The Griffith criterion is the condition of rock microcrack expansion rather than macroscopic destruction, which is suitable for analyzing the loading conditions of interpenetrating the coal seam without damaging the coal seam. A fissure of length a, the critical stress at the tip of the tear fissure, is characterized by the following equation [25]:

$$\sigma_c = \left(\frac{2E\gamma}{\pi a}\right)^{1/2} \tag{1}$$

In the formula, $\gamma$ represents the surface free energy per unit area of material, namely the energy required for material forming per unit of crack area; E represents the elastic modulus of coal.

The literature [26] shows the change of surface energy with the temperature in coal bodies with different structures (Figure 5). The surface energy of structural coal is between 10 and 30 mJ/m$^2$, and the elastic modulus of most tectonic coal is less than 1 GPa. With this data as a constraint, the density and sound velocity of the tectonic coal are 1300 kg/m$^3$ and 2000 m/s [27]. According to Equation (2), the stress required to expand the cracks of different lengths in different surface energy seams, and the sound pressure level, sound strength and sound strength level corresponding to the stress can be calculated

(Tables 1 and 2). This stress value is significantly less than the tensile strength of the coal seam, so it will not damage the coal seam macroscopically.

$$I = P^2/2\rho C \tag{2}$$

In the formula, $P$ is sound pressure, kPa; $\rho$ is density, kg/m$^3$; $C$ is sound speed, m/s.

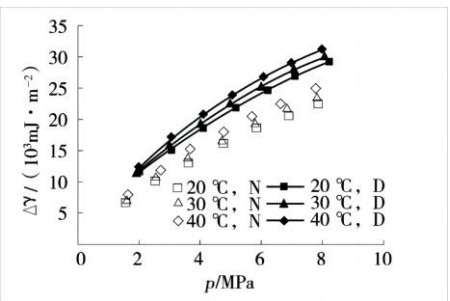

N, Native structural coal; D, Tectonic coal

**Figure 5.** Change of surface energy of native structural and tectonic coals with temperature; N, native structural coal; D, tectonic coal [26].

**Table 1.** Crack extended stress at fissure length of 1 cm in tectonic coal seam.

| Surface Energy (J/m²) | Modulus of Elasticity (GPa) | | | | | | | | | | | |
| | 0.2 | | | | 0.5 | | | | 0.8 | | | |
| | Stress Value (kPa) | Pressure Level (dB) | Sound Intensity (kW/m²) | Sound Intensity Level (dB) | Stress Value (kPa) | Pressure Level (dB) | Sound Intensity (kW/m²) | Sound Intensity Level (dB) | Stress Value (kPa) | Pressure Level (dB) | Sound Intensity (kW/m²) | Sound Intensity Level (dB) |
|---|---|---|---|---|---|---|---|---|---|---|---|---|
| 0.01 | 11.3 | 175 | $2.46 \times 10^{-2}$ | 134 | 17.8 | 179 | $6.1 \times 10^{-2}$ | 138 | 22.6 | 181 | $9.82 \times 10^{-2}$ | 140 |
| 0.02 | 16 | 178 | $4.92 \times 10^{-2}$ | 137 | 25.2 | 182 | 0.122 | 141 | 31.9 | 184 | 0.196 | 143 |
| 0.03 | 19.5 | 180 | $7.31 \times 10^{-2}$ | 139 | 30.9 | 184 | 0.184 | 143 | 39.1 | 186 | 0.294 | 145 |

**Table 2.** Crack extended stress at fissure length of 1 mm in tectonic coal seam.

| Surface Energy (J/m²) | Modulus of Elasticity (GPa) | | | | | | | | | | | |
| | 0.2 | | | | 0.5 | | | | 0.8 | | | |
| | Stress Value (kPa) | Pressure Level (dB) | Sound Intensity (kW/m²) | Sound Intensity Level (dB) | Stress Value (kPa) | Pressure Level (dB) | Sound Intensity (kW/m²) | Sound Intensity Level (dB) | Stress Value (kPa) | Pressure Level (dB) | Sound Intensity (kW/m²) | Sound Intensity Level (dB) |
|---|---|---|---|---|---|---|---|---|---|---|---|---|
| 0.01 | 36 | 185 | 0.249 | 144 | 56 | 189 | 0.6 | 148 | 70 | 191 | 0.942 | 150 |
| 0.02 | 50 | 188 | 0.48 | 147 | 80 | 192 | 1.23 | 151 | 100 | 194 | 1.92 | 153 |
| 0.03 | 62 | 190 | 0.739 | 149 | 98 | 194 | 1.85 | 153 | 124 | 196 | 2.96 | 155 |

### 2.3. Simulation Experiment of Coal Seam with High-Strength Acoustic Wave Excitation and Permeability Enhancement

Based on the acoustic oil production research, the former Soviet Union found that the sound field strength required to start the heat transfer process of the sandstone reservoir was 1 kW/m² (150 dB) [28]. In the late 1990s, academician Xian Xuefu proposed the idea of using the acoustic shock method to improve the gas extraction rate, and established the gas adsorption/desorption model and seepage theory under the action of ground stress field, temperature field and electric field [29–34].

Researchers in China conducted physical simulation experiments on coal samples with a diameter of 50 mm and a length of 100 mm under various conditions using

a 40 kHz frequency and a 30 W power acoustic source [8,35–37]. The study found that under the average effective stress condition of 4 MPa, the applied sound field can improve the permeability of the coal sample, reflecting that the sound wave connects the pores and fissures in the coal seam in tearing mode (Figure 6). With the extension of the acoustic loading time, the coal sample permeability gradually increases (Figure 7). At the same time, it was found that, under the action of sound waves, the gas desorption law is unchanged, but the desorption amount increases (Figure 8). After the sound field operation, the adsorption isothermal curve of coal remains unchanged, but the adsorption capacity decreased significantly (Figure 9).

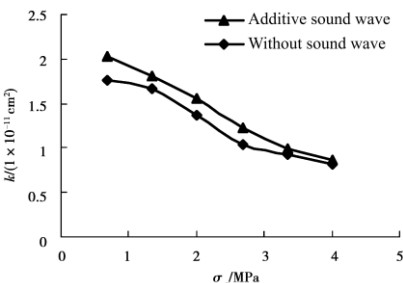

**Figure 6.** Relationship between coal permeability and average effective stress under acoustic field.

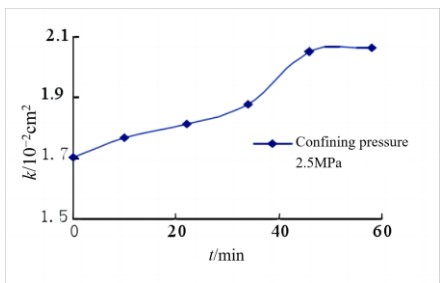

**Figure 7.** Relation between time of loading acoustic wave and coal permeability.

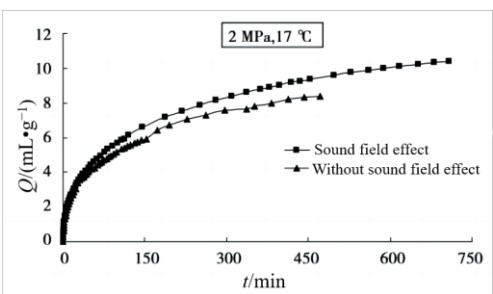

**Figure 8.** Relationship between gas desorption amount and loading sound field time.

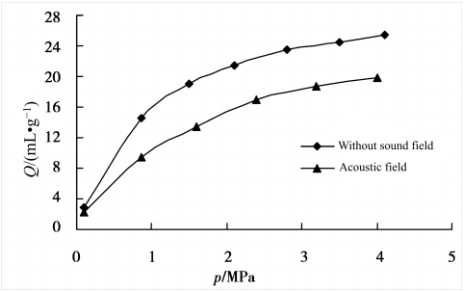

**Figure 9.** Isothermal adsorption curves under loading sound waves.

Shi Qingmin (2018) studied the crack evolution characteristics of three samples of fat coal, lean coal and anthracite under high-frequency low-sound intensity (50 kHz, 6 kW/m$^2$ (158 dB)), high-frequency high-sound intensity (50 kHz, 10 kW/m$^2$ (160 dB)) and low-frequency high-sound intensity (20 kHz, 10 kW/m$^2$ (160 dB)) [38]. The results show that under the action of a sound field, the surface fissure and internal fissure of coal rock have the characteristics of simultaneous expansion and interconnection, which eventually lead to the fragmentation of coal rock (Figure 10).

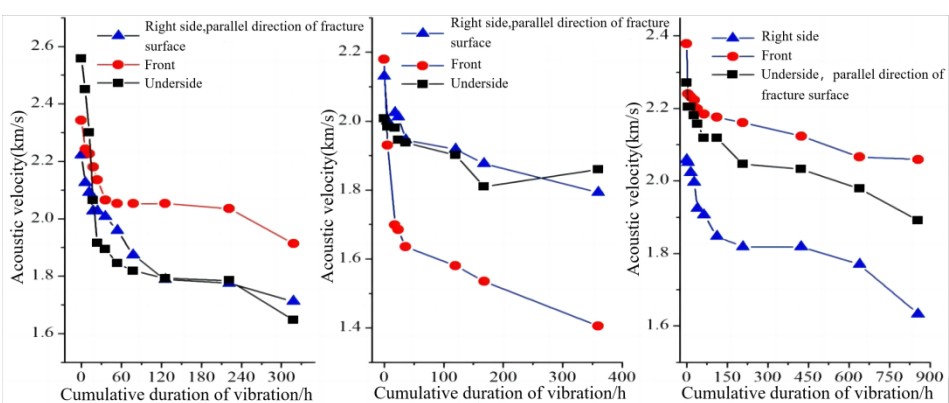

**Figure 10.** Wave velocity response of fracture evolution in coal and rock under ultrasonic field.

Due to the lack of low-frequency strong sound sources, domestic and foreign studies have been conducted using high-frequency acoustic sources. Although the sound frequencies are different, the results are basically the same. Only the result of very low frequency vibration below 30 Hz showed opposite findings [39]. According to the actual measurement results of acoustic transmission in the oil industry, only acoustic waves below 200 Hz can propagate more than 100 m in the sandstone layer [40]. For this reason, a 100~200 Hz sound wave is selected for calculation, and referring to the above experimental results, the sound intensity threshold of the more permeable coal seam is set at 5 kW/m$^2$.

### 2.4. Field Test of High-Strength Sound Wave Increases the Coal Seam

Academician Aici Qiu's team analyzed the results of the completed pilot test [41–44], and believed that the controlled shock waves were coupled to the coal seam with the help of a water medium; the coal seam is not only the object of shock wave operation, but also the medium of propagating shock wave. The released energy successively forms the shock wave band, compression wave band and elastic wave band in the coal seam. This is through rupture, tearing, high-strength elastic wave disturbance and other modes [16] to improve the permeability of the coal seam and promote the gas desorption [45,46].

The repetitive application of shock waves can further induce the fatigue effect on the mechanical properties of the coal seam [47]. Loading it with relatively low energy can make the coal seam crack growth, pore rupture and form the fissure network [48], thus enhancing the coal seam permeability. Effective communication pores and cracks, are conducive to the coal bed adsorption gas desorption, diffusion and seepage. In addition, the single-point repeated operation with low energy can ensure the overall integrity of the coal seam structure and prevent the hole wall breakage and collapse. It was found that the shock waves loaded in the coalbed drilling holes and coalbed methane wells can attenuate into strong sound waves in the coal seam, continue to crack the coal seam and promote gas adsorption and seepage.

An enhanced permeability shock wave was applied during drilling operations in the structural coal seam of the Yushe Coal Mine in Shuicheng, Guizhou Province. The shock wave had a strength of 80 MPa when applied to the borehole wall. On one side of a working face, a row of parallel boreholes was drilled at a distance of 40 m from each other (Figure 11). Shock wave operations were carried out in different boreholes, which changed

the gas extraction flow rate attenuation mode and greatly improved the efficiency of gas extraction (Figure 12) [16]. Based on the analysis of monitoring drilling flow without shock wave operation, it was found that the effective action radius of high-strength sound wave is more than 40 m.

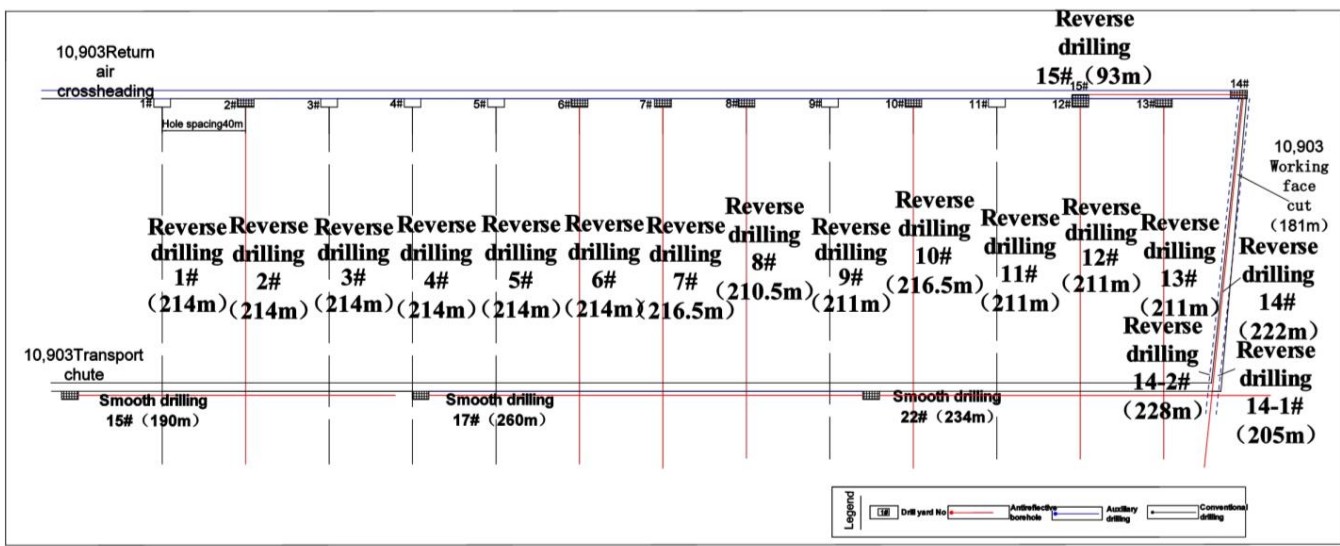

**Figure 11.** Layout of coal seam gas drainage project in 10,903 working faces.

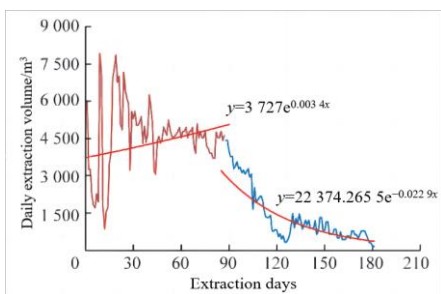

**Figure 12.** Gas flow curves and attenuation trend in typical boreholes.

Theoretical analysis, experimental simulation and field practice have demonstrated that the upper limit threshold of a high-strength acoustic wave, calculated based on the coal seam's tensile and shear strength, and the lower threshold of acoustic wave intensity required for coal seam fracture propagation, calculated based on the Griffith fracture theory, ensure that high-intensity sound waves can expand cracks in the coal seam without causing macroscopic damage to the coal seam. For structural coal seams, when high-intensity sound waves of 5 kw/m$^2$ (157 db)~10 kw/m$^2$ (160 db) are applied to the coal seams, the fractures of tectonic coal seams can be expanded and gas seepage can be promoted.

## 3. Implementation Method of High-Strength Sound Wave Increasing Coal Seam

At present, the most effective measure to increase the coal seam is to mine the protective layer and realize the penetration of the protective layer through the pressure discharge of the protective layer. However, a variety of penetration measures implemented in this coal seam have not effectively solved the problem of structural coal seam gas. Drawing on the idea of protective layer mining and penetrating the protective layer, the implementation area of the shock wave increasing the coal seam is transferred from the coal seam to the top and bottom slab layer. On the one hand, the problem that the structural coal seam is difficult to form a hole, and on the other hand, the top and bottom plate can become a large area sound source to produce high-strength sound waves in situ.

### 3.1. In-Situ Generating Method of High-Strength Sound Waves

The method of high-strength sound wave generation should first abandon the technical route of using instruments and equipment to produce sound waves and then loading it to the coal seam. In other words, according to the principle of acoustic wave generation, the top floor of the coal seam is used as an energy converter to impact the top floor, and the attenuation and evolution of the impact and vibration of the roof floor is used to produce the high-strength sound wave required for the penetration in the coal seam in situ.

When the shock wave is loaded on the roof and floor rock layer, the impact directly breaks the rock layer in the area where the impact strength is higher than the compressive strength of the rock layer and attenuates due to energy consumption. In the area where the impact strength is lower than the compressive strength of the rock layer, but higher than the tensile and shear strength of the rock layer, the rock layer continues to crack in the mode of tension and tear. In the area where the impact strength is lower than the tensile and shear strength of rock stratum, the impact attenuation evolves into high-strength sound wave, and then enters the coal seam to expand the original crack (Figure 13).

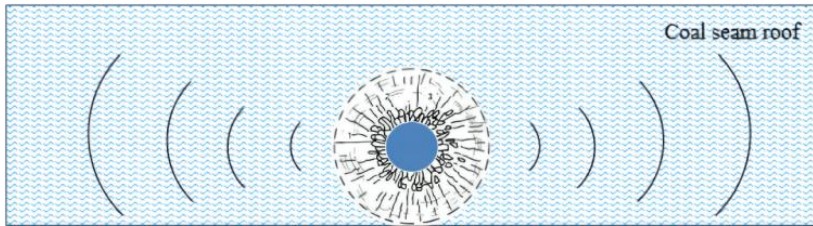

**Figure 13.** Impact effect of the controlled shock wave in coal seam roof.

### 3.2. Implementation Scheme Design

(1)　Design parameters

The boreholes in the roof floor can be drilled from the coal seam roadway or on the ground. The distance between the impact point and the coal seam in the borehole is designed based on the mechanical strength of the overlying and underlying rock layers, the attenuation law of shock waves in the rock layers and the drilling technology. This ensures that the shock wave intensity at the interface between the rock layer and the coal seam is just below the rock's tensile and shear strength after the shock wave does work in the rock layer due to attenuation, reducing the damage to the coal seam caused by the shock wave. For the structural coal seam with high outburst risk, the shock wave strength should be attenuated below the tensile strength of the coal seam, so that the high-strength sound wave entering the coal seam does not have the effect of damaging the coal seam structure.

The roof and floor of the coal seam can be drilled from the coal seam roadway or on the ground. According to the mechanical strength of the roof and floor rock layer, the attenuation law of the shock wave in the rock layer and the drilling technology, the distance between the impact point in the borehole and the coal seam is designed so that the strength of the shock wave after attenuation in the rock layer is just less than the tensile and shear strength of the rock layer at the interface between the rock layer and the coal seam, reducing the damage of the shock wave to the coal seam. For the tectonic coal seam with a high outburst risk, the shock wave strength should be attenuated below the tensile strength of the coal seam so that the high-strength sound wave entering the coal seam does not damage the coal seam structure.

The propagation law of blast shock wave in rock formation is described by the following formula [49]:

$$P_r = P_0 \left(\frac{r_0}{r}\right)^n \tag{3}$$

In the formula, $P_r$ represents the peak pressure value at any point from the source, kPa; $P_0$ represents the peak pressure value generated by the source, kPa; $r_0$ represents the radius of the source, cm; r represents the distance between the observation point and the

source, cm; n represents constant, n = 2 ± μ/(1 − μ); μ represents the Poisson's ratio of rock. The fracture zone with shock wave amplitude greater than the compressive strength of rock stratum is taken as "+", and the fracture zone with a shock wave amplitude lower than the compressive strength of rock stratum but higher than the tensile shear strength of rock stratum is taken as "−".

Taking the shock wave peak 200 MPa and the $C_{7+8}$ coal seam thin siltstone and siltstone direct roof as an example, the average compressive strength of saturated water rock is 23 MPa, the average tensile strength of natural dry rock is 1.23 MPa, the density is 2460 kg/m$^3$, the Poisson ratio is 0.2 and the sound velocity is 3000 m/s. Therefore, according to the calculation of formula (3), the radius of the rupture area caused by a single shock wave is 146 cm, that is, the source area radius of the elastic wave is 146 cm, the sound pressure is 1.23 MPa and the corresponding sound strength is $1 \times 10^5$ W/m$^2$ (170 dB). At this time, it is equivalent to placing a spherical high-strength sound source with a radius of 146 cm in the roof and floor rock. Considering that the tensile strength of tectonic coal is 0.5 MPa (sound pressure level is 208 db and sound intensity is $4.17 \times 10^4$ w/m$^2$ (sound intensity level 166 db)), and the tearing radius of a single shock wave is calculated to be 245 cm, which is more favorable for the protection of coal seam macrostructures.

(2)    Design of perforhole arrangement

The upstream/downstream drilling is arranged in the coal seam bottom rock roadway (Figure 14), with large diameter (133 mm) air drilling to improve the percentage of drilling in the structural coal seam. In order to avoid the impact damage to the coal seam, the operation point is selected in the top and bottom slab layer in the drilling hole, and the distance between the working point and the top and bottom plate and the interface of the coal seam is the distance between the shock wave attenuation and the tensile shear strength of the coal seam. For sandstone roof and bottom bed and a tensile shear strength of 1 MPa, this distance is designed to be between 152 and 245 cm.

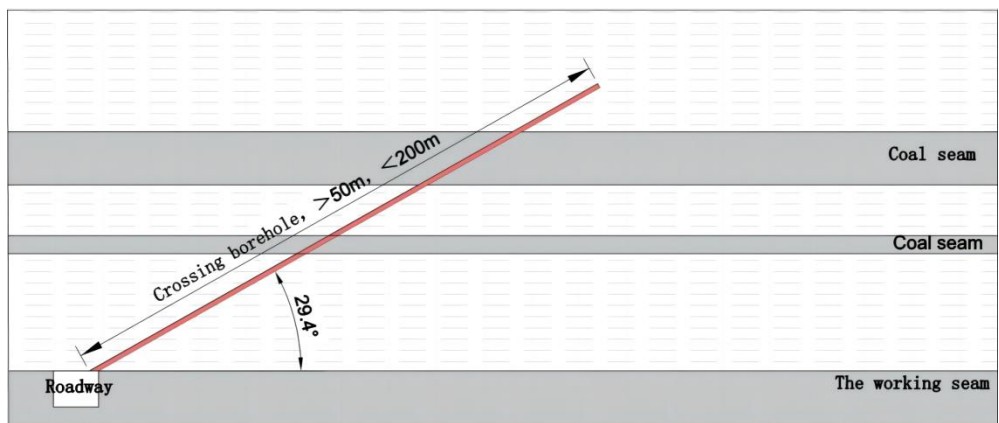

**Figure 14.** Schematic diagram of controlled shock wave in stacked coal seam drilling hole.

(3)    Design of horizontal hole operation layout in heading face

In response to the difficult problems of the poor pre-pumping effect of drilling at the cover digging face of the prominent coal seam and the difficulty of regulating the contradiction between digging and pumping, a stepped drill field is designed in the roadway heading face, and the long-distance directional horizontal drilling is implemented in the drill field under the seam roof/floor (Figure 15).

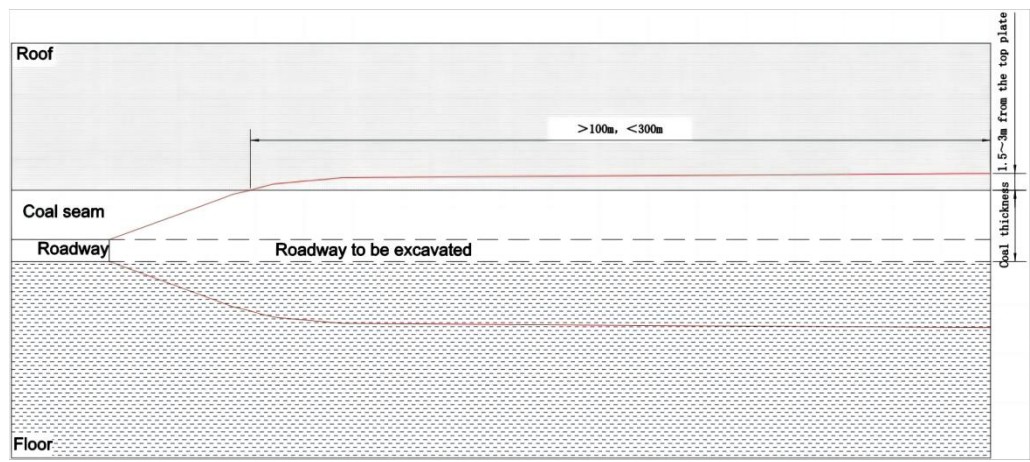

**Figure 15.** Schematic diagram of directional drilling layout of controllable shock wave in covering roadway tunneling.

To ensure that fissures are generated in the rock mass to connect the coal seam and the drilling operation point, the distance between the horizontal borehole and the interface of the coal seam/overlying and underlying rock layers is the distance at which the shock wave attenuates to the tensile and shear strength of the rock mass in the rock layer. For a typical sandstone roof, the distance is less than 146 cm. Through the establishment of a fracture channel connected with the coal seam through the fractured rock layer, the high-strength sound wave attenuated and evolved by the fractured rock layer directly enters the coal seam, realizing the purpose of protecting the coal seam to be excavated and eliminating the risk of local outburst.

(4)    Design of the channel roof drilling operation layout scheme

In order to meet the engineering requirements for efficient pre-pumping at the working face of this coal seam, the roof or bottom plate drilling is arranged in the groove on both sides of the working face, and the pre-pumping of the coal seam gas is implemented (Figure 16). Taking the above typical sandstone as an example, the distance between the interface of the roof downstream borehole and the coal seam is 146 cm.

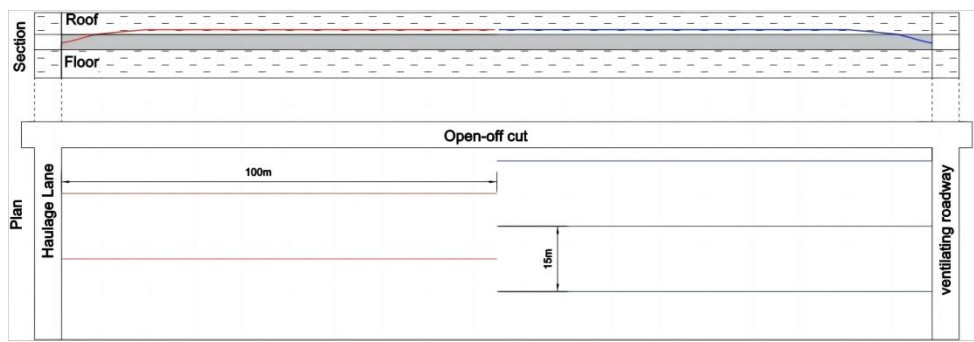

**Figure 16.** Layout of controllable shock wave in coal seam cover tunneling and pre-pumping drilling of working surface.

### 3.3. Prediction of the Effective Action Range of the High-Strength Sound Wave

Estimate the effective transmission distance of high-intensity sound waves generated by shock wave evolution. The propagation rules of sound intensity and the sound pressure of cylindrical elastic sound wave in rock strata are as follows [49]:

$$J(r) = J_0 \sqrt{\frac{r_0}{r}} e^{-\alpha(r-r_0)} \tag{4}$$

$$P(r) = P_0 \sqrt{\frac{r_0}{r}} e^{-\alpha(r-r_0)} \tag{5}$$

In the formula, $J(r)$ represents the distance from the shaft in the rock formation or coal seam where the sound intensity is at $r$, kw/m$^2$; $P(r)$ represents the distance from the shaft in the rock formation or coal seam where the sound pressure is at $r$, MPa; $J_0$ and $P_0$ are the sound intensity and acoustic pressure at the interface where the shock wave decay affects the elastic acoustic wave, rather than the power and peak pressure of the shock wave source, kw/m$^2$, MPa; $r$ represents the distance between any point in the rock bed or coal seam and the shaft, m; $r_0$ represents the radius of the source which is the distance from the interface of the shock wave attenuation to the elastic sound wave; m;$\alpha$ represents the attenuation coefficient of the shock wave in the rock seam or the coal seam, m$^{-1}$.The measured attenuation coefficient of acoustic waves in sandstone is shown in Table 3 [40].

**Table 3.** Attenuation coefficient of different frequency sound waves in sandstone reservoirs.

| Frequency of sound wave $f$ (Hz) | 10 | 50 | 100 | 200 | 500 | 1000 | 20,000 |
|---|---|---|---|---|---|---|---|
| Attenuation coefficient $\alpha$ (m$^{-1}$) | 0.00268 | 0.013 | 0.025 | 0.046 | 0.134 | 0.28 | 6.85 |

**Frequency of sound wave:** The number of vibrations per second of the sound source, expressed as $f$ (frequency); **Attenuation coefficient:** Attenuation coefficient is also called attenuation constant. It is the real part of the propagation coefficient. It consists of two parts: classical absorption and molecular absorption.

According to the design, the impact strength attenuates to the top and shear strength at the interface between the top and bottom plates. Taking the tensile shear strength of the top and bottom plate as the sound pressure value of the high-strength sound wave, the transmission distance of the high-strength sound wave in the coal seam is calculated according to Equations (4) and (5). According to the test results of Shi Qingmin [38], the area with a sound intensity value higher than 5 kW/m$^2$ is regarded as the effective action area of a high-intensity sound wave. Taking the mechanical parameters of the roof layer of the coal seam of Bailongshan Mine as an example, the sound intensity of 5.9 kW/m$^2$ (152 dB) is loaded from the roof sandstone to the coal seam in a 30 m radius. The high-strength acoustic source area reaches 30 m, and the area directly applied to the coal seam is a circular area with a radius of 30 m (Table 4).

**Table 4.** Intensity of sound waves at different frequencies with propagation distance (kW/m$^2$).

| Distance | Frequency | |
|---|---|---|
| | **100 Hz** | **200 Hz** |
| 10 m | 30.9 (165 dB) | 12.8 (164 dB) |
| 20 m | 25.8 (164 dB) | 11.5 (161 dB) |
| 30 m | 10.8 (160 dB) | 5.9 (158 dB) |
| 40 m | 7.3 (159 dB) | 3.2 (155 dB) |
| 50 m | 5.1 (157 dB) | 1.8 (153 dB) |

## 4. Conclusions

1. Based on theoretical analysis, experimental results and a field test, the theoretical and technical parameters of a high-intensity acoustic wave used to increase the penetration of coal seams are proposed. The method of impact and conversion into a high-strength sound wave in the roof and bottom plate solves the problem of hole forming in the structural coal seam, and it avoids the problem of the outburst of coal and gas caused by the impact of the structural coal seam.

2. The implementation method is proposed to implement the impact at the top and bottom of the penetration hole and use the attenuation effect of the top and bottom

to generate high intensity acoustic waves to increase coal seam penetration. That is, in the top or bottom construction drilling, the strong shock wave cracks and tears the roof and bottom plate to communicate the coal seam. The top and bottom plate is used to convert the shock wave into a high-strength sound wave into the structural coal seam, and the top and bottom impact drilling is also used as a gas extraction drilling.

3. According to the output parameters of the existing shock wave equipment to produce peak 200 MPa, the effective range of a strong impact to crack and tear typical sandstone, and the parameters and effective range of strong shock wave evolution into a high-strength sound wave are estimated to provide a basis for the design of the drilling position.

4. Based on theoretical analysis and test results, the implementation method is not limited to controllable shock wave technology, but can also use the existing deep hole pre-crack loose blasting, carbon dioxide cracking as sound source. However, relevant operational parameters need to be re-estimated and the safety management of the roof should be ensured.

**Author Contributions:** Formal analysis, Y.Z. (Youzhi Zhao) and S.Z.; Investigation, D.W. and Y.L.; Resources, C.C.; Data curation, J.R.; Writing—original draft, W.L.; Writing—review & editing, Y.Z. (Yongmin Zhang) and Y.Q. All authors have read and agreed to the published version of the manuscript.

**Funding:** This research was funded by [Huaneng Group Headquarters] grant number [HNKJ21-HF09 Experimental Study on Controllable Shockwave Presplitting of Hard Top Coal in Liuxiang Coal Mine].

**Conflicts of Interest:** The authors declare no conflict of interest.

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
