# Peer review of "Theoretical Basis and Technical Method of Permeability Enhancement of Tectonic Coal Seam by High Intensity Acoustic Wave In Situ"

_processes, doi:10.3390/pr11082372_

Round 1

Reviewer 1 Report

Figure:  Enhancement of all figures is strongly needed. Some were so blurred and barely seen. Please use higher resolution.

pg 5, para 2 : "Griffith The criterion is...." . Please check the phrase and wording.

pg 8, Para 1 :  "both tdomestic and foreign" ; typos?

Please check the typos. i.e.: spacing between the last word and new line

pg 9, para 2 :  "Academician Qiu Aici's team analyzed the field test results,..." ; please provides necessary references of this work as this seem important to support the theoretical framework proposed by the authors of this manuscript. 

References: Please check all formatting.

The quality of English language needs some revision. 

Author Response

  1. Higher resolution, clearer images have been used.
        Page 5, paragraph 2: "Griffith, the standard is ......" Revise to "The Griffith standard is ......" .

  2.     Page 8, first paragraph: "Domestic and international" was revised to read "Domestic and international research has been conducted using high frequency sound sources".
  3.     Page 9, paragraph 2: "Academician Qiu Aiqi's team analyzed the results of the field tests and..." New references related to the tests were provided.

Reviewer 2 Report

This manuscript introduces a novel concept for improving coal seam permeability in tectonically deformed coal seams, where conventional boreholes face challenges due to poor borehole wall stability. The proposed method involves implementing coal seam permeability enhancement engineering in the surrounding rock using shock wave technology. Through a series of experiments, including physical simulations and on-site tests, the paper demonstrates the significant impact of high-strength acoustic waves in enhancing coal seam permeability. The core of the proposed approach lies in loading shock waves on the roof or bottom rocks of coal seams to create cracks that can communicate with the coal seams. As these shock waves attenuate into high-strength sound waves within the rock layers before entering the coal seams, this technique provides a novel and efficient solution for extracting gas from tectonically deformed coal seams at a reduced cost.

This study is very interesting and attractive to related researchers. After some minor revision, the paper can be published. The comments are listed below:

1.       While the primary focus of this paper is on the attenuation of shock waves into high-strength sound waves in the roof and bottom rocks, it is worth noting that the vibration induced by shock waves also contributes to the transformation of shock waves into sound waves. Therefore, it is suggested that the author further explores this aspect in future research to enrich the understanding and applicability of this innovative approach.

2.       The literature review in the introduction requires improvement, as it lacks recent studies on the relevant subjects. To enhance the context and strengthen the manuscript, it is crucial to incorporate representative references that encompass recent research findings.

3.       The English language proficiency in this document requires improvement. The language used throughout the manuscript would greatly benefit from refinement and optimization to achieve a more professional standard.

See above.

Author Response

New references have been added